# Omics-Inferred Partitioning and Expression of Diverse Biogeochemical Functions in a Low-O₂ Cyanobacterial Mat Community

Sharon L. Grim,[a]* Alexander A. Voorhies,[a] Bopaiah A. Biddanda,[b] Sunit Jain,[a] Stephen C. Nold,[c] Russ Green,[d]§ Gregory J. Dick[a,e]

[a]Department of Earth and Environmental Sciences, University of Michigan, Ann Arbor, Michigan, USA
[b]Annis Water Resources Institute, Grand Valley State University, Muskegon, Michigan, USA
[c]Biology Department, University of Wisconsin—Stout, Menomonie, Wisconsin, USA
[d]Thunder Bay National Marine Sanctuary, National Oceanic and Atmospheric Administration, Alpena, Michigan, USA
[e]Cooperative Institute for Great Lakes Research, University of Michigan, Ann Arbor, Michigan, USA

Sharon L. Grim and Alexander A. Voorhies contributed equally to this work. The order was determined alphabetically by last name.

**ABSTRACT** Cyanobacterial mats profoundly influenced Earth's biological and geochemical evolution and still play important ecological roles in the modern world. However, the biogeochemical functioning of cyanobacterial mats under persistent low-O₂ conditions, which dominated their evolutionary history, is not well understood. To investigate how different metabolic and biogeochemical functions are partitioned among community members, we conducted metagenomics and metatranscriptomics on cyanobacterial mats in the low-O₂, sulfidic Middle Island sinkhole (MIS) in Lake Huron. Metagenomic assembly and binning yielded 144 draft metagenome assembled genomes, including 61 of medium quality or better, and the dominant cyanobacteria and numerous *Proteobacteria* involved in sulfur cycling. Strains of a *Phormidium autumnale*-like cyanobacterium dominated the metagenome and metatranscriptome. Transcripts for the photosynthetic reaction core genes *psaA* and *psbA* were abundant in both day and night. Multiple types of *psbA* genes were expressed from each cyanobacterium, and the dominant *psbA* transcripts were from an atypical microaerobic type of D1 protein from *Phormidium*. Further, cyanobacterial transcripts for photosystem I genes were more abundant than those for photosystem II, and two types of *Phormidium* sulfide quinone reductase were recovered, consistent with anoxygenic photosynthesis via photosystem I in the presence of sulfide. Transcripts indicate active sulfur oxidation and reduction within the cyanobacterial mat, predominately by *Gammaproteobacteria* and *Deltaproteobacteria*, respectively. Overall, these genomic and transcriptomic results link specific microbial groups to metabolic processes that underpin primary production and biogeochemical cycling in a low-O₂ cyanobacterial mat and suggest mechanisms for tightly coupled cycling of oxygen and sulfur compounds in the mat ecosystem.

**IMPORTANCE** Cyanobacterial mats are dense communities of microorganisms that contain photosynthetic cyanobacteria along with a host of other bacterial species that play important yet still poorly understood roles in this ecosystem. Although such cyanobacterial mats were critical agents of Earth's biological and chemical evolution through geological time, little is known about how they function under the low-oxygen conditions that characterized most of their natural history. Here, we performed sequencing of the DNA and RNA of modern cyanobacterial mat communities under low-oxygen and sulfur-rich conditions from the Middle Island sinkhole in Lake Huron. The results reveal the organisms and metabolic pathways that are responsible for both oxygen-producing and non-oxygen-producing photosynthesis as well as

Address correspondence to Gregory J. Dick, gdick@umich.edu.

*Present address: Sharon L. Grim, NASA Ames Research Center, Moffett Field, California, USA.

§Present address: Russ Green, Office of National Marine Sanctuaries, National Oceanic and Atmospheric Administration, Sheboygan, Wisconsin, USA.

The authors declare no conflict of interest.

interconversions of sulfur that likely shape how much $O_2$ is produced in such ecosystems. These findings indicate tight metabolic reactions between community members that help to explain the limited the amount of $O_2$ produced in cyanobacterial mat ecosystems.

**KEYWORDS** cyanobacteria, geomicrobiology, metagenomics, metatranscriptomics, photosynthesis, biogeochemistry, mats, oxygen, sulfur

Cyanobacterial mats host communities of microorganisms that are linked through metabolic interactions in which the products of one metabolism are the substrate for another (1–4). These metabolic interactions underpinned critical biogeochemical processes throughout Earth's history (5–7) and continue to do so in the modern world (2, 4). Cyanobacterial mats have been a prevalent feature of the biosphere for billions of years and strongly influenced the composition of the atmosphere (7, 8). Most prominently, cyanobacteria mediated the oxygenation of Earth's surface by producing $O_2$ via oxygenic photosynthesis, thus catalyzing a cascade of geobiological transitions that set the stage for complex life (9).

Modern microbial mats have long served as analogs for studying their ancient equivalents, and recent work has made great progress in elucidating cyanobacterial mat processes, organisms, and their interactions (10–12). However, relatively little work has been devoted to cyanobacterial mats that inhabit persistently low-$O_2$ and/or sulfidic environments. This is a critical gap in knowledge, because low-$O_2$, sulfidic phototrophic habitats were likely common in the Precambrian (13) and thus prevailed for much of the evolutionary history of cyanobacteria (6, 14). Further, cyanobacteria were likely anoxygenic phototrophs prior to evolving oxygenic photosynthesis (15–18), and anoxygenic cyanobacteria may have delayed Earth's oxygenation during ~2 billion years of low-$O_2$ conditions in the Proterozoic (6, 19, 20).

Sulfide is a key control of the physiology of cyanobacteria and the biogeochemical cycling of elements in cyanobacterial mats (6). Cyanobacteria typically conduct oxygenic photosynthesis, which is inhibited by sulfide because it blocks photosystem II (PSII) (21). However, some cyanobacteria can tolerate sulfide through a variety of mechanisms, including sulfide-resistant oxygenic photosynthesis, simultaneous operation of oxygenic and anoxygenic photosynthesis, and a complete switch to anoxygenic photosynthesis using sulfide as the electron donor (21). In some strains, sulfide can either inhibit or enhance oxygenic photosynthesis, depending on light availability and sulfide conditions (22). Sulfide-quinone reductase (SQR) is the key enzyme for anoxygenic photosynthesis by cyanobacteria; it oxidizes sulfide and transfers electrons to PSI through the quinone pool, effectively bypassing PSII (23–25). SQR is a diverse protein family that has also been linked to sulfide detoxification in cyanobacteria and other phototrophs (6, 24). Although studies have elucidated the physiological responses of cyanobacteria to sulfide and the role of SQR in anoxygenic photosynthesis (21, 26, 27), little is known about transcriptomic controls on cyanobacterial anoxygenic photosynthesis within cyanobacterial mats.

The Middle Island sinkhole (MIS) in Lake Huron, MI, hosts cyanobacterial mats in low-$O_2$, intermittently sulfidic conditions (28). The mats sit atop anoxic, organic-rich sediments in which microbial methanogenesis and sulfate reduction produce methane and sulfide, leading to sharp redox gradients (29–32). The mats are metabolically versatile, having the ability to conduct oxygenic photosynthesis, anoxygenic photosynthesis, and chemosynthesis (30, 33, 34). Despite this metabolic versatility, early 16S rRNA gene and metagenomic studies suggested that the mats have low taxonomic diversity, being dominated by just one cyanobacterial genotype, an organism closely related to *Phormidium autumnale* (29, 30, 34). However, deep 16S rRNA gene sequencing of the mat and underlying sediments revealed a taxonomically diverse microbial community, including numerous groups of sulfate-reducing and sulfur-oxidizing bacteria that are suggested to mediate key biogeochemical processes within and beneath the mat (31).

Further, diurnal vertical migration of sulfur-oxidizing bacteria and diatoms exerts a strong influence on the biogeochemistry of the systems and on light availability and photosynthesis in the mats (33, 35).

In order to investigate how different metabolic and geochemical functions are partitioned among community members and expressed over time, we conducted metagenomic analysis on 15 samples collected at seven time points between 2007 and 2012 and metatranscriptomic analysis on six samples taken during day and night in 2012. The *Phormidium* species was found to dominate transcriptional activity in the MIS mat community and displays gene expression patterns consistent with a mixture of oxygenic and anoxygenic photosynthesis. We also recovered genomes and transcripts of diatoms, sulfate-reducing bacteria, and sulfur-oxidizing bacteria, providing insights into the microbial groups that mediate key biogeochemical processes within the mat.

## RESULTS AND DISCUSSION

**Environmental setting and conditions.** The environmental and geological setting of the MIS was described previously (28). In May 2012, at the time of collection of samples for metatranscriptomic and metagenomic sequencing (Table S1), the groundwater layer ~1 m immediately above the mat in the sinkhole had substantially elevated specific conductivity (1,813 $\mu$S cm$^{-1}$ versus 226 $\mu$S cm$^{-1}$ in the ambient lake water), lower and temporally consistent temperature (7 to 9°C), and an average dissolved O$_2$ level of 3.37 mg L$^{-1}$.

**Community composition and function.** Assembly and binning produced 16 high-quality draft metagenome-assembled genomes (MAGs) (>90% completion, <5% redundancy), 45 medium-quality (>50% completion, <10% redundancy), and 79 low-quality draft MAGs (<50% completion, <10% redundancy) according to estimates based on single-copy genes expected to be present (36) (Table S2). In addition, four MAGs had high redundancy (>10%), including three of the most abundant MAGs (Bin_4_1, Bin_1, Bin_235_243; *Rhodoferax*, *Phormidium*, and *Planktothrix*, respectively), which had high coverage and moderate completion (Table S2). For example, the dominant MAG in most samples, *Phormidium* (Bin_1), had high redundancy (56%) and moderate completeness (70%). Single-copy genes in the *Phormidium* MAG were on small contigs, consistent with fragmentation of contigs due to high coverage and strain heterogeneity (37, 38), and they were classified taxonomically as various cyanobacteria, as expected based on the lack of available *Phormidium* genomes (Fig. S1). Thus, these MAGs likely contain contigs from multiple strains of *Phormidium*.

While community membership was dynamic across time and space, *Phormidium* was consistently the dominant organism in the MIS mats (Fig. S2). Other cyanobacteria were also abundant in the mat, including *Planktothrix* (formerly referred to as *Oscillatoria* in previous studies of MIS, but its 16S rRNA genes are most similar to those of *Planktothrix agardhii* and *Planktothrix rubescens* [39]), *Pseudanabaena*, and *Spirulina*. MAGs were also recovered for various *Bacteroidetes*, *Betaproteobacteria*, *Chloroflexi*, *Deltaproteobacteria*, *Epsilonproteobacteria*, *Firmicutes*, *Gammaproteobacteria*, and *Spirochaetes* (Table S2). In most cases there were multiple MAGs recovered for each of these taxonomic groups. Many of these groups are commonly found in anoxic or hypoxic sediments (40, 41), and several are enriched in sediments below the mats at the MIS (29, 31).

To investigate which community members have metabolic pathways for mat biogeochemical processes, we searched the MAGs for key genes involved in carbon metabolism, nitrogen and sulfur cycling, oxygenic and anoxygenic photosynthesis, and other energy metabolisms. Cyanobacteria were the dominant phototrophs in terms of genomic abundance; *Phormidium* had a mean genomic coverage of over 200×, though based on high redundancy (56%), multiple strains are present (Table S2). While two putative diatom MAGs (Bin_3_1 and Bin_3_3) had low average coverages (0.31 and 6.19×), their chloroplasts were very abundant (up to 230× coverage), likely reflecting their high copy number per cell and easier assembly than the nuclear genome (Fig. S3). Marker genes of anoxygenic photosynthesis, including photosynthetic reaction center (*pufM* and *pufL*) and bacteriochlorophyll synthesis (*bchB* and *bchL*), were

present in *Rhodoferax* (Bin_4_1) and *Chloroflexi* (Bin_120) MAGs (Table S2). Diatoms and *Chloroflexi* are often associated with cyanobacterial mats; migratory diatoms play important roles in nitrogen cycling in MIS mats and sediments (35), and *Chloroflexi* engage in tight metabolic interactions with cyanobacteria (2, 3, 10, 42).

The MAGs of various proteobacteria revealed organisms involved in sulfur cycling. Key genes for dissimilatory sulfite reductase (*dsrA*) involved in sulfate reduction were present in deltaproteobacterial genomic bins, including one unclassified *Desulfobacteraceae* member, one unclassifiable *Desulfobulbaceae* member, and one *Desulfobacula* member (Table S2). Based on their relatively high abundance, we infer that these sulfate-reducing bacteria were present directly in the cyanobacterial mat (43, 44) rather than the alternative that the sequences could be due to contamination of the mat by underlying sediments. Related sulfate-reducing bacteria are associated with anoxygenic bacteria in Lake Mahoney (45), with cyanobacterial mats at Guerrero Negro (46, 47), and in nearby mats of chemolithotrophic sulfur oxidizers influenced by the same groundwater as MIS (48).

Potential for oxidation of elemental sulfur using reverse dissimilatory sulfite reductase (*rdsrA*) was detected in genomic bins of *Arcobacter* (*Epsilonproteobacteria*), several *Betaproteobacteria*, *Thiothrix*, and *Thioploca* and in unbinned scaffolds putatively belonging to *Beggiatoa* (Table S2). *Thiothrix*, *Thioploca*, and *Beggiatoa* are likely the white filamentous bacteria observed directly underneath the cyanobacterial mat (29) and in a nearby artesian fountain fed by the same groundwater (48). They can migrate on diel cycles and influence the balance of oxygenic versus anoxygenic photosynthesis by modulating light available to phototrophs when covering the mat (33). These large sulfur-oxidizing bacteria likely contribute to the substantial rates of chemosynthesis measured previously (30) and likely influence cyanobacterial photosynthesis by consuming sulfide. Potential for thiosulfate oxidation, indicated by the presence of *soxA*, was observed in betaproteobacterial, deltaproteobacterial, and gammaproteobacterial bins (Table S2). Finally, the *mmoC* gene, for methane oxidation, was identified within a MAG classified as *Methylococcales* (Table S2).

**Whole-community transcriptomics.** Metatranscriptomic sequencing was conducted to investigate the *in situ* metabolic activity of the MIS mat community members. Although transcript abundance is not directly proportional to protein abundance or enzymatic activity, transcriptomics provides valuable insights into which community members and metabolic pathways are active at the time of sampling and their response to environmental conditions (49). In order to evaluate the influence of light availability on gene expression in MIS mats, three samples collected in 2012 at 1 p.m. and three collected at 1 a.m. were studied. In terms of relative abundance, transcripts mapped to MAGs from *Phormidium*, *Bacteroidetes*, *Thiotricaceae*, and the putative diatom dominated the metatranscriptome (Fig. S2; Table S2). Other significant contributors (>1× mean coverage) to the transcript pool were bins from *Paludibacter* and other *Bacteroidetes* members, *Rhodoferax* (*Betaproteobacteria*), *Chloroflexi*, *Planktothrix*, and a variety of unidentified bins (Table S2). Mapping of metatranscriptomic data to marker genes and MAGs provided a picture of the organisms responsible for metabolic/biogeochemical processes within the mat in day and night (Fig. 1; Table S2).

**Transcripts involved in phototrophy.** We next focused our transcriptomic analysis on key genes for photosynthesis. Core components of the reaction centers of PSI and PSII, encoded by *psaA* and *psbA* genes, respectively, are degraded at an enhanced rate compared to other proteins due to absorption of excess light energy from photosynthesis (50, 51). This leads to higher cellular demand for protein and likely explains the high abundance of transcripts we observed for these genes. The most abundant transcripts for *psaA* and *psbA* genes were from *Phormidium* and the diatom, with minor contributions from *Planktothrix*, *Spirulina*, and *Pseudanabaena* (Fig. 2). Included in our analyses were multiple versions of the cyanobacterial *psbA* genes, encoding the D1 subunit of PSII, which are expressed according to light and redox conditions (52, 53) and have been suggested to be involved in sulfide tolerance and/or anoxygenic photosynthesis in cyanobacteria (54). *Phormidium* contained three of the four *psbA* types, and type 3 had the most transcripts (Fig. 2). This type of *psbA* is expressed during

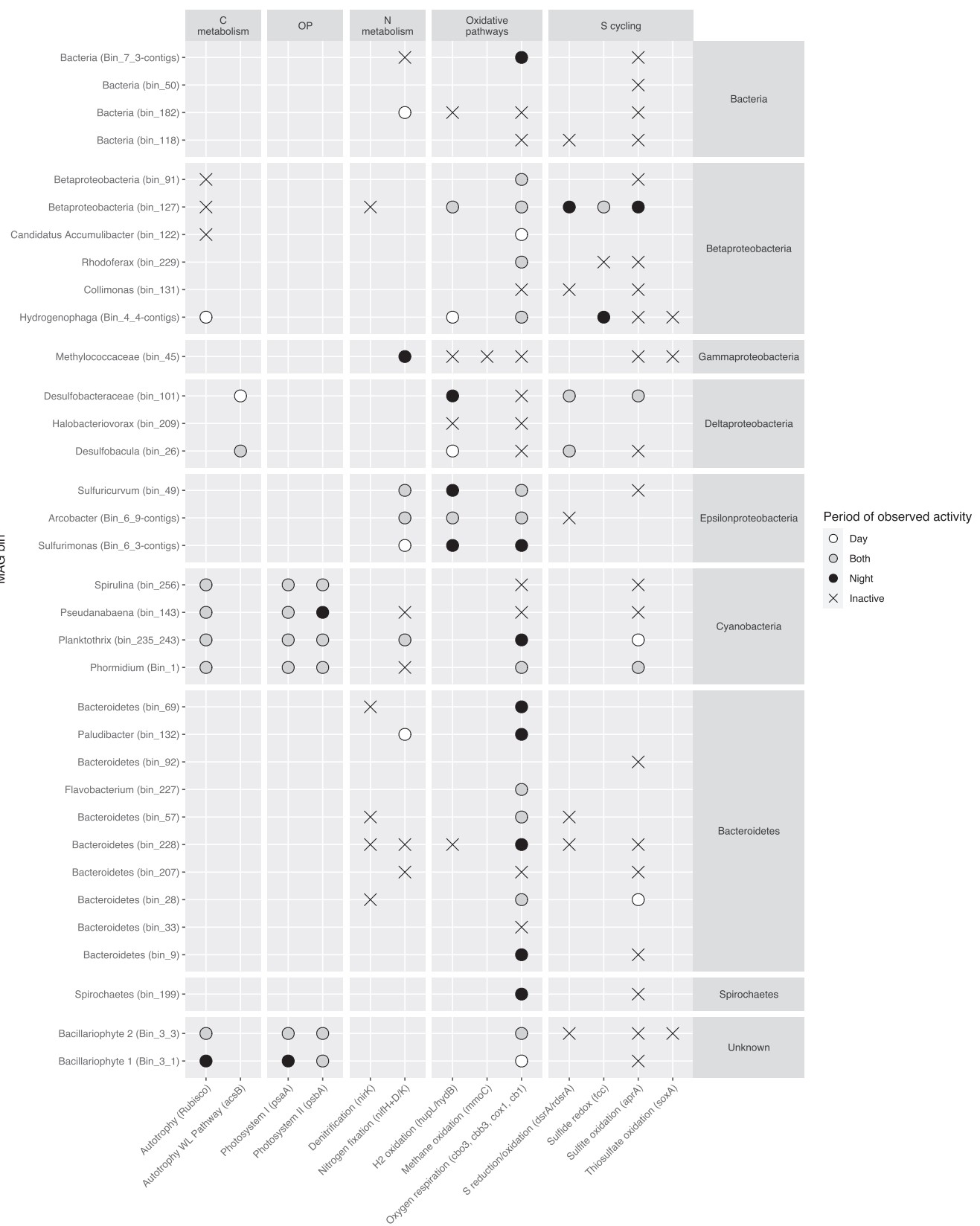

**FIG 1** Detection of transcripts from marker genes of key metabolic/biogeochemical processes in MAGs (see Table S2 for details). Symbols are colored according to the time of day at which transcripts were detected: white, day; black, night; gray, both night and day. "X" indicates that the gene was observed in the MAG, but no transcripts were detected.

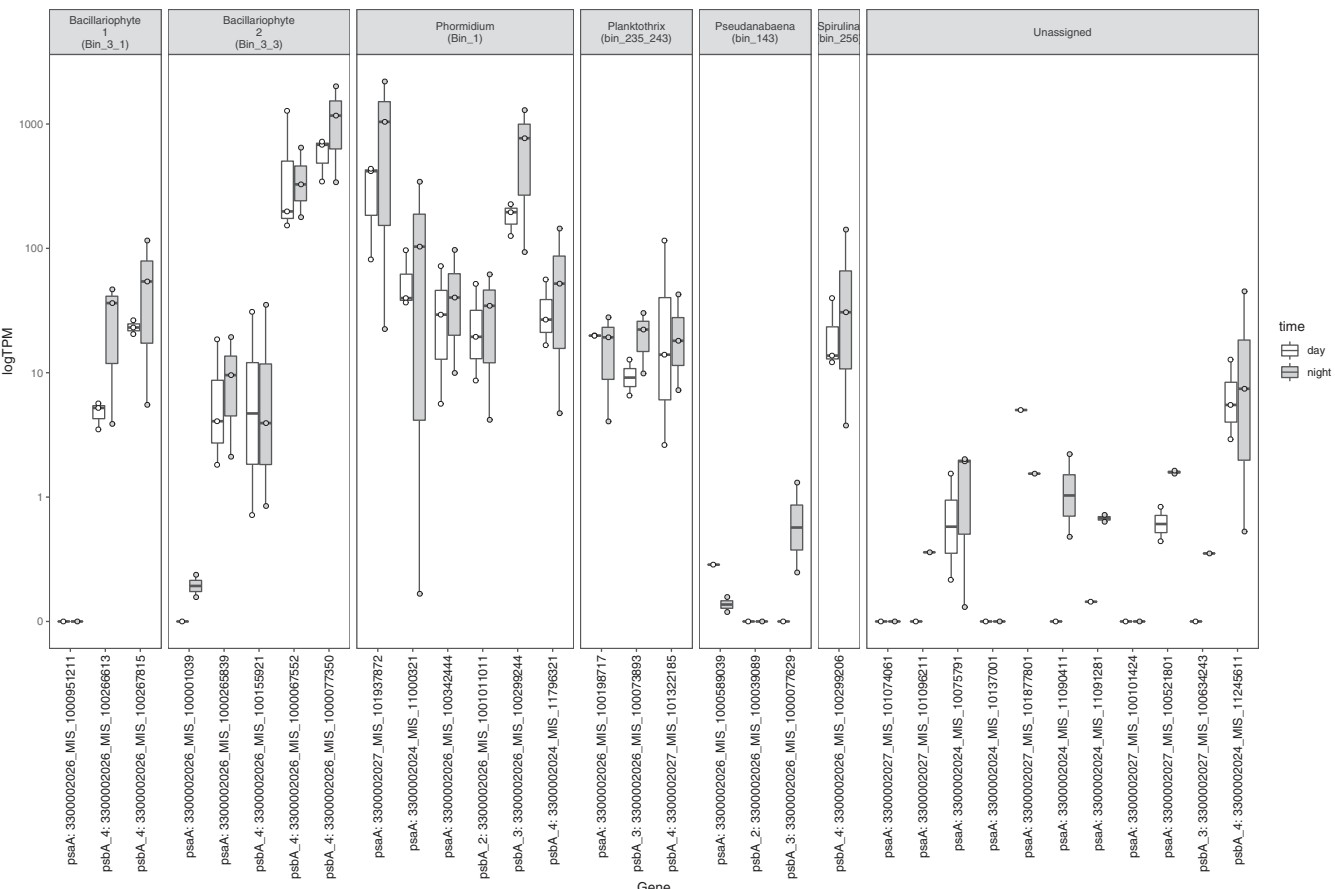

**FIG 2** Relative abundance of transcripts for photosystem genes normalized by total number of sequences in each sample. Log-transformed transcript abundance in the day (white) and night (gray) of genes encoding photosystem I (*psaA*) and photosystem II (*psbA*) is shown for each MAG (top) with box-and-whisker plots. Boxes represent the 25th to 75th percentiles, the inside line indicates the median, and whiskers extend to minimum and maximum values. Observations are overlaid as points. The *x*-axis labels "psbA_2," "psbA_3," and "psbA_4" refer to *psbA* types (see the text).

microaerobic and/or dynamic redox conditions (55, 56). The other cyanobacterium that was abundant in metagenomic data sets, *Planktothrix*, had type 3 and 4 *psbA* genes, with similar relative abundance of transcripts in day and night but at much lower levels than *Phormidium*.

Twenty-one of 32 *psaA* and *psbA* genes had transcripts that were more abundant at night than in the day, although these differences were not statistically significant (Fig. 2). When normalized by the number of transcript reads recruited to each bin, which removes effects of transcriptomic variability across the whole community on transcript counts for genes within each bin, 9 of 21 genes had transcripts that were more abundant at night (Fig. S4). These patterns contrast those in several laboratory studies of cyanobacterial transcription, which found highest expression of photosynthesis reaction core genes during the day (57–62). Several field studies have also shown highest expression of PSII genes during the day (63–65). One possible explanation is that whereas many previous studies focused on oxygenic unicellular cyanobacteria that typically undergo rapid cell division (59, 66), *Phormidium* species are filamentous and typically much slower growing (0.07 to 0.5 day$^{-1}$, depending on light and nutrient availability) (67, 68). There is a precedent for high transcription of photosynthetic genes in the dark and lower transcription in the light; in *Synechococcus* sp. strain PCC 7002, transcript levels of several *psbA* genes were constant across several conditions, including light and dark (69).

Phototrophic genes were among the most highly expressed genes in marine surface waters collected 3 h before sunrise (70). Photosystem I genes in thrombolites were constitutively expressed, with nearly even transcript abundance at midday and

mSystems®

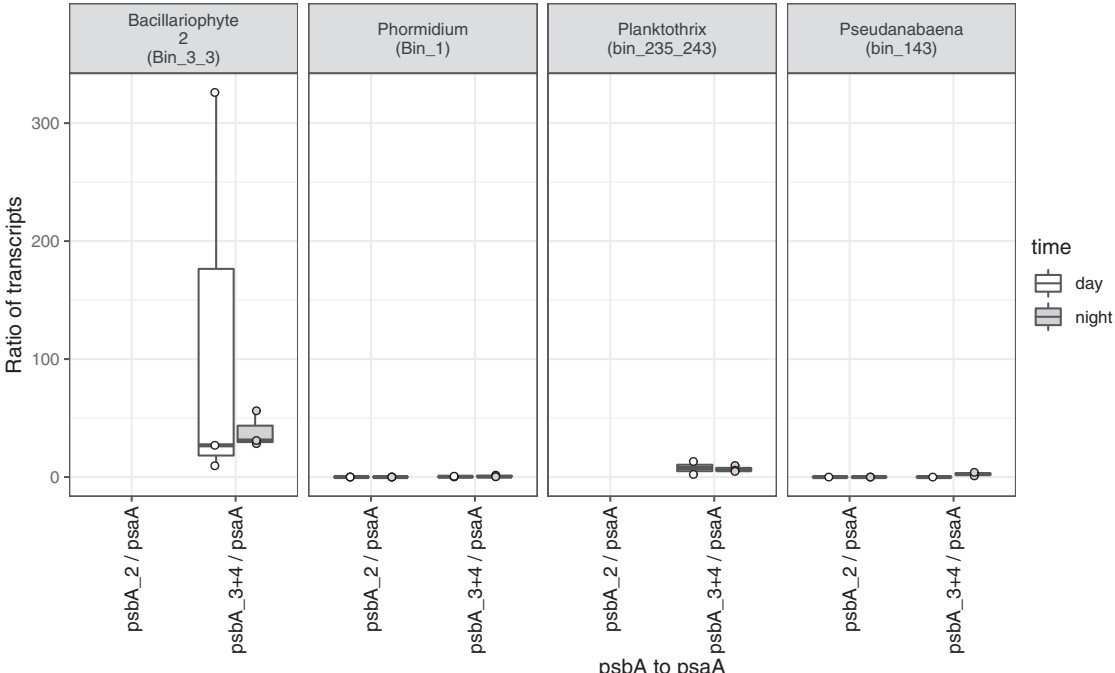

**FIG 3** Ratio of transcript abundances for photosystem II (*psbA*) to photosystem I (*psaA*) genes in day and night, shown for each MAG (top) with box-and-whisker plots. Boxes represent the 25th to 75th percentiles, the inside line indicates the median, and whiskers extend to minimum and maximum values. Observations are overlaid as points.

midnight (64). Anoxygenic phototrophs such as *Chloroflexi* and *Chlorobi* also express structural components of the photosynthetic apparatus at night (11, 71). Finally, our results could also be explained in part by decreased rates of afternoon photosynthesis that have been observed in cyanobacterial mats, including sharp drops at midday (72–74), which have been attributed to limitation of dissolved inorganic carbon (2). It should also be noted that transcript levels do not necessarily reflect protein abundance; under some circumstances, *psbA* messages accumulate without synthesis of the D1:2 protein (75, 76).

The abundance of transcripts obtained from *Phormidium* PSII genes indicates that genes for oxygenic photosynthesis were transcriptionally active at the time of sampling. However, *Phormidium* had on average more than two times higher abundance of transcripts for PSI genes than PSII genes (*psbA*-to-*psaA* ratio < 0.5) (Fig. 3). In contrast, the diatom chloroplast recruited more than 25 times more transcripts to PSII genes (*psbA*) than PSI genes (*psaA*). However, likely due to high variability of the abundance of diatom transcripts for these genes, this large difference in the ratio of transcript abundance from PSII and PSI genes was not statistically significant.

Other MIS cyanobacteria exhibited an intermediate ratio, with 2 to 7 times more PSII than PSI transcripts. A high ratio of PSII to PSI transcripts was also found in *Prochlorococcus*, a unicellular marine planktonic cyanobacterium (59). We infer that the higher relative abundance of PSI transcripts in *Phormidium* (and perhaps other MIS cyanobacteria) reflects transcriptional regulation, either via downregulation of PSII genes or upregulation of PSI genes, to conduct anoxygenic photosynthesis in the presence of sulfide. Although to our knowledge these are the first transcriptional data from anoxygenic cyanobacteria, they are consistent with the physiological shift toward PSII-independent anoxygenic photosynthesis that was reported previously (21, 27), with a decrease in the stoichiometry of PSII-PSI in response to sulfide (77), and with genes for anoxygenic photosynthesis being inducible via transcriptional regulation (24, 78, 79). The stoichiometry of PSII-PSI can also be regulated according to light levels (80); the PSII/PSI ratio is lower at lower light levels, which also favors anoxygenic photosynthesis in the MIS mat system (33).

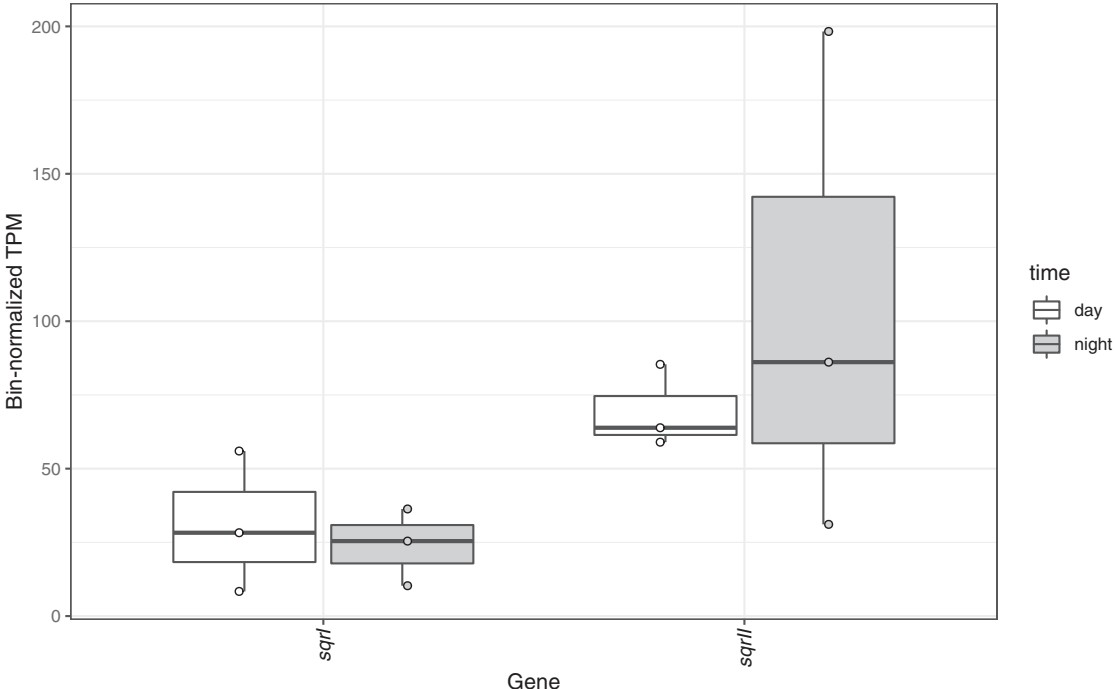

**FIG 4** Relative abundance of transcripts for sulfide quinone reductase (SQR) type I (putatively involved in anoxygenic photosynthesis) and type II (putatively involved in sulfide detoxification) genes from the *Phormidium* MAG, normalized by transcripts recruited to the MAG for each sample. Boxes represent the 25th to 75th percentiles, the inside line indicates the median, and whiskers extend to minimum and maximum values. Observations are overlaid as points.

Sulfide quinone oxidoreductase (SQR) transfers electrons from sulfide to PSI during anoxygenic photosynthesis (24). Of the five cyanobacterial *sqr* homologs recovered from the MIS community, the *Phormidium* SQR had the highest abundance of transcripts, from both its type I and II *sqr* genes (Fig. 4). The bin-normalized transcripts-per-million (TPM) value of *Phormidium*'s type II SQR was significantly higher than that for SQRI ($P < 0.05$). While MAGs of *Planktothrix* and *Pseudanabaena* have SQRs (6), transcripts for these genes were not observed. The *Phormidium* SQRs showed transcript abundance comparable to that of the PSI genes *psaL* and *psaX* (Fig. S5). Little is known about how anoxygenic photosynthesis and sulfide tolerance are regulated at the genetic level in cyanobacteria. Expression of type II SQR for sulfide detoxification in *Synechocystis* sp. strain PCC6803 (79) and type I SQR for anoxygenic photosynthesis in *Geitlerinema* sp. strain 9228 (24, 81, 82) is inducible by sulfide. Both constitutive expression (25, 83, 84) and inducible expression (85, 86) of *sqr* have been observed in anoxygenic bacteria. There was little metatranscriptomic evidence of anoxygenic photosynthesis by anoxygenic bacteria (i.e., *Chloroflexi* or *Betaproteobacteria*). Genes for photosynthetic reaction cores (*pufM* and *pufL*) and bacteriochlorophyll (*bchB* and *bchL*) were not highly expressed, with 0 or 1 read mapped in all samples. Overall, these results suggest that the cyanobacteria are largely responsible for anoxygenic photosynthesis previously measured in MIS mats (30, 33).

The most highly expressed terminal oxidase for respiration in *Phormidium* was a cytochrome *bd*-type oxidase (Fig. S6), which has exceptionally high affinity for $O_2$, with a $K_m$ of 3 to 8 nM (87). The high transcriptional activity of this low-$O_2$ respiratory oxidase is consistent with adaptation to low-$O_2$ conditions for extended time periods.

**Transcripts involved in sulfur cycling and carbon fixation.** Transcripts of seven different *dsrA* genes were observed, and the presence of *dsrD* on the same scaffold (Fig. S7) was used to confirm inclusion of these genes in the dissimilatory sulfite reductase pathway (*dsr* genes). *dsrD* is useful a marker of sulfite reduction because it is absent from organisms that use homologous *rdsrA* genes for sulfur oxidation (88, 89). Transcriptionally active reductive *dsrA* genes were present in seven MAGs representing

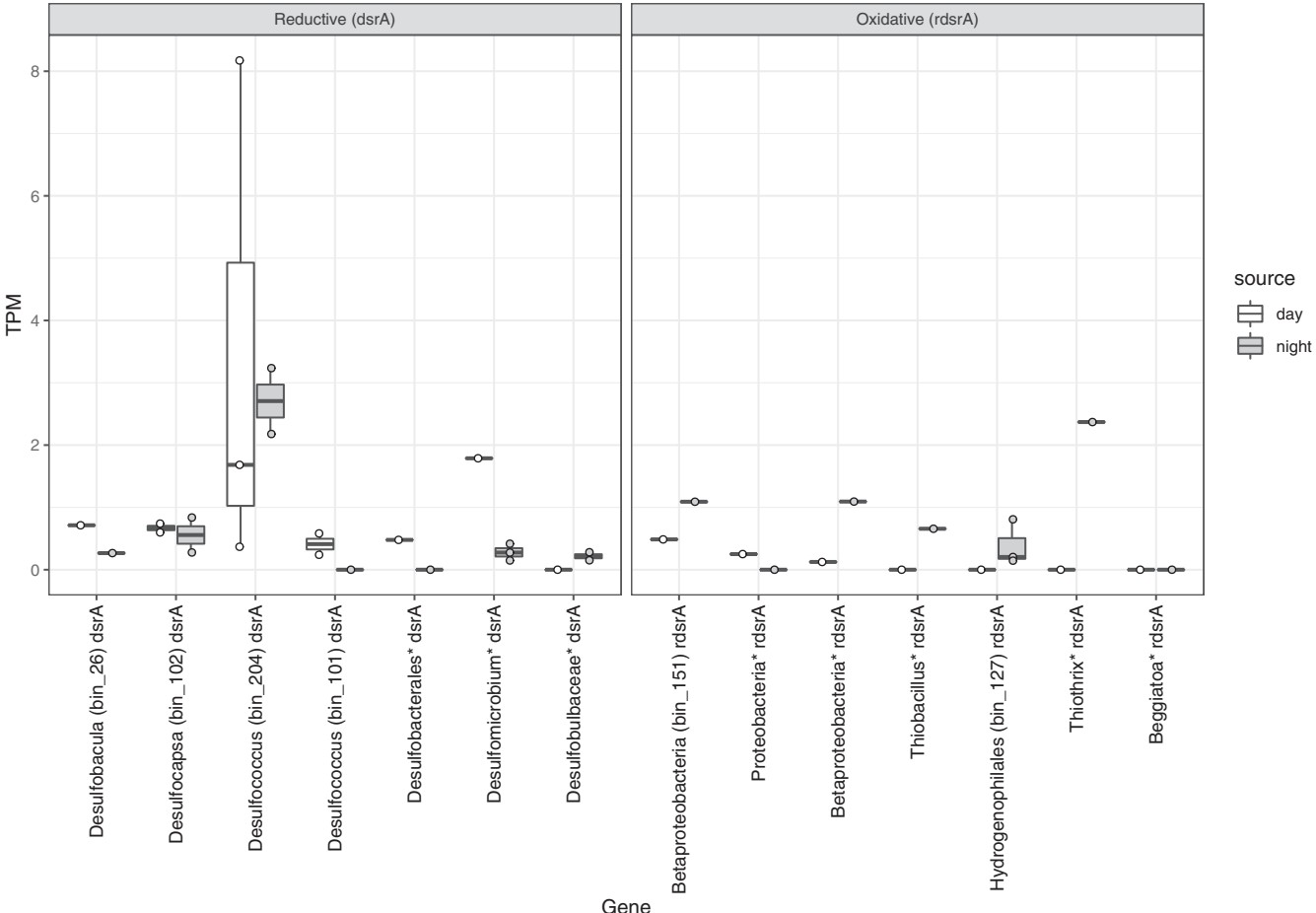

**FIG 5** Relative abundance of transcripts for key genes for dissimilatory sulfite reduction (*dsrA*) and reverse dissimilatory sulfite reduction (*rdsrA*) normalized by total number of sequences in each sample. Boxes represent the 25th to 75th percentiles, the inside line indicates the median, and whiskers extend to minimum and maximum values. Observations are overlaid as points.

six genera of *Deltaproteobacteria*, with most transcripts coming from *Desulfococcus* (*Deltaproteobacteria*; *Desulfobacterales*), followed by *Desulfomicrobium* (Fig. 5). These results reveal the organisms responsible for active sulfate reduction within the cyanobacterial mat, which has been measured at high rates by $^{35}SO_4^{2-}$ tracer studies (32). For most sulfate-reducing MAGs, *dsrA* transcripts were detected and even more abundant during the day, suggesting sulfate reduction during the photosynthetic period and likely metabolic interactions with cyanobacteria via the cycling of sulfur and/or carbon (6, 90). These sulfate-reducing bacteria are also present in sediments underlying the cyanobacterial mat (31).

Seven *rdsrA* genes for sulfur oxidation were observed, including those in MAGs from three *Betaproteobacteria* (two unclassified and one member of the *Hydrogenophilales*), one classified only as proteobacteria, and one *Thiobacillus* organism (Fig. 5). Two unbinned genes most similar to *Thiothrix* and *Beggiatoa* (*Gammaproteobacteria*) were also recovered. With the exception of the unbinned proteobacterial gene, all of these *rdsrA* genes had more transcripts at night. Transcripts from *soxA* genes for thiosulfate oxidation were detected, with those from *Rhodoferax* (*Betaproteobacteria*) and unbinned representatives of the *Campylobacterales* (*Epsilonproteobacteria*) having the highest abundance of transcripts (Fig. S8). Transcripts of genes for sulfide oxidation via flavocytochrome *c* sulfide dehydrogenase (*fcc*) were also observed in bins from the *Betaproteobacteria* and *Gammaproteobacteria*, though their sample-normalized transcript abundance was nearly an order of magnitude lower than those of *dsrA*.

To assess sources of primary production at MIS, we measured abundance of transcripts encoding key genes of four autotrophic pathways: ribulose-1,5-bisphosphate carboxylase/

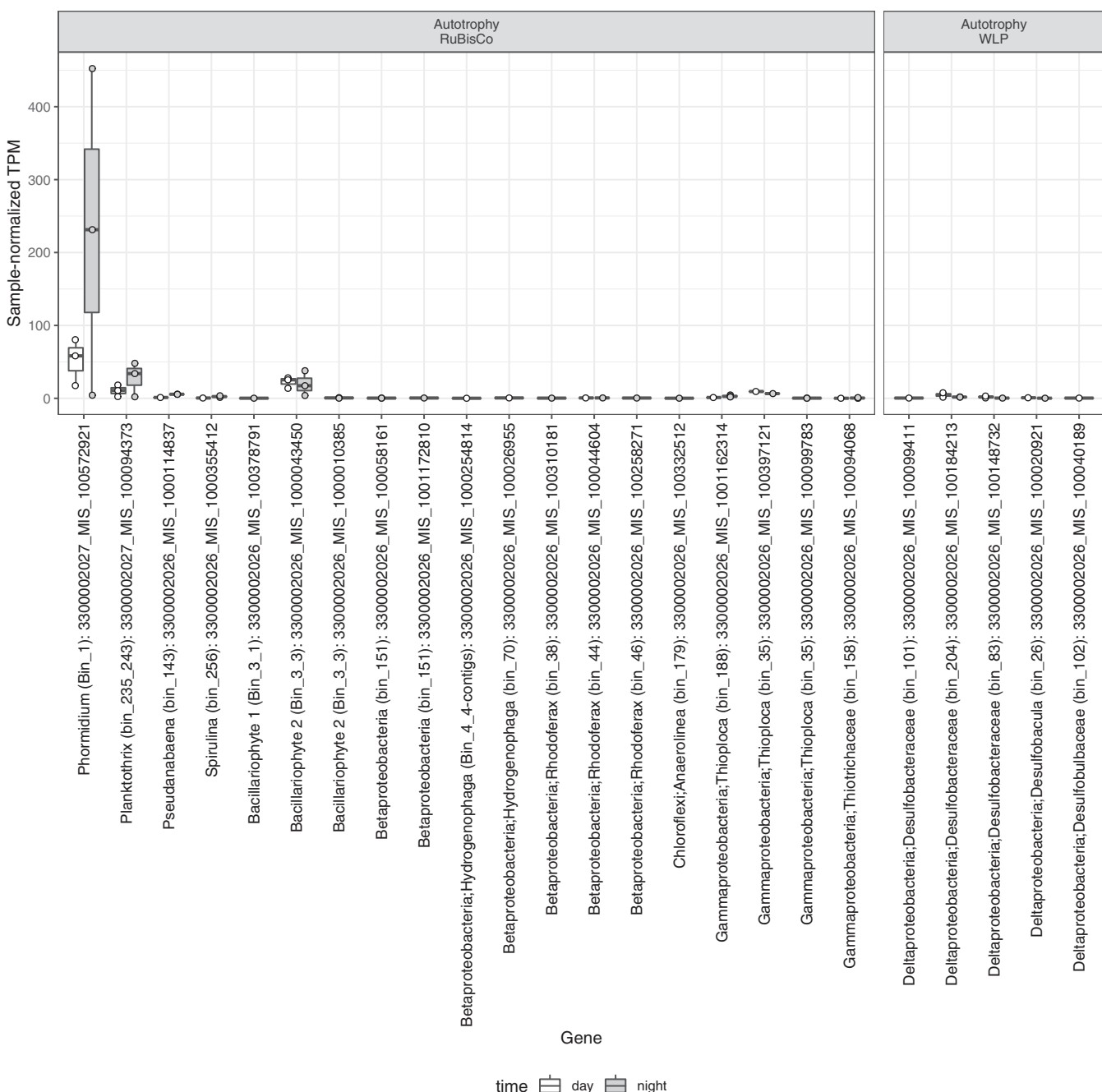

**FIG 6** Relative abundance of transcripts for key genes in active autotrophic pathways normalized by total number of sequences in each sample. The sample-normalized TPM of each gene in daytime (white) and nighttime (gray) are plotted as box and whiskers in which boxes represent the 25th to 75th percentiles, the inside line indicates the median, and whiskers extend to minimum and maximum values. Observations are overlaid as points. Bin and taxonomy of indicator genes for RuBisCO (*rbcL*) and the Wood-Ljungdahl pathway (*acsB*) are indicated on the x axis.

oxygenase (RuBisCO) for the Calvin cycle (*rbcL*), ATP citrate lyase (*aclB*) for the reverse tricarboxylic acid cycle, CO dehydrogenase/acetyl coenzyme A (acetyl-CoA) synthase (*acsB*) for the Wood-Ljungdahl pathway, and malyl-CoA/(S)-citramalyl-CoA lyase (*mcl*), malonyl-CoA reductase/3-hydroxypropionate dehydrogenase, and 3-hydroxypropionyl-CoA dehydratase (*mcr*) for the 3-hydroxypropionate cycle (Fig. 6). RuBisCO had the highest transcript abundance of any autotrophic pathway. *Phormidium*, *Planktothrix*, and a diatom chloroplast actively transcribed *rbcL*, and the cyanobacteria especially were active at night. *Thiotricaceae* also expressed *rbcL* but at substantially lower levels than the phototrophs. Several *Desulfobacterales* (*Deltaproteobacteria*) MAGs expressed the Wood-Ljungdahl genes, with higher expression

during the day. We did not observe transcriptional activity of *aclB*, *mcl*, or *mcr* genes in any samples.

**Conclusions.** Our data on taxon-specific genomic content and gene expression patterns provide insights into the microbial players and pathways that mediate biogeochemistry in anoxygenic cyanobacterial mats, which likely underpinned critical aspects of Earth's geobiological evolution but have been understudied in the modern world. The metatranscriptomic data revealed that a *Phormidium autumnale*-like cyanobacterium previously found to be the dominant community member (29, 30) is also responsible for the majority of transcripts for photosynthesis, underscoring its essential role in the MIS ecosystem. Taken together, low expression ratios of PSII genes to PSI genes and expression of SQR indicate that *Phormidium* within MIS mats conducts anoxygenic photosynthesis with sulfide as the electron donor, consistent with previous geochemical measurements at the MIS (30, 33) as well as studies of other members of the *Oscillatoriales* (24, 91). Hence, the *Phormidium* population appears to be metabolically versatile, capable of both oxygenic and anoxygenic photosynthesis. It is not clear whether this phototrophic versatility stems from niche adaptation among closely related ecotypes (e.g., differential activity of strains that are oxygenic and anoxygenic specialists) or true cellular versatility in which *Phormidium* cells switch pathways depending on sulfide concentration. Further, our bulk sampling of mats was not sensitive to the vertical microgradients of sulfide concentration, so we are unable to evaluate potential vertical stratification of oxygenic/anoxygenic photosynthesis within the mat.

This study provides a picture of how metabolisms encoding specific biogeochemical functions are partitioned among mat community members (Fig. 1). Primary production occurs via oxygenic photosynthesis by cyanobacteria and diatoms, anoxygenic photosynthesis by cyanobacteria (with minor contributions potentially from *Chloroflexi* and *Betaproteobacteria*), and chemosynthesis via sulfur oxidation by *Thiotrichales* and *Chromatiales* (*Gammaproteobacteria*) and *Epsilonproteobacteria*. Sulfide sets the stage for these metabolisms and is produced via sulfate and/or sulfur reduction by several different groups of *Deltaproteobacteria* both within the cyanobacterial mat and in underlying sediments. Bacteroidetes are the major heterotrophs, consuming organic carbon released as photosynthetic exudate or via viral lysis (92). Although the spatial arrangement of these processes within the mat remains unresolved, sulfide, oxygen, and organic carbon are likely rapidly cycled between the organisms producing and consuming them (1–4). Such tightly coupled interactions would help explain why there is little mat-derived carbon sequestered in the sediments (93). Tight coupling of O$_2$ production and consumption metabolisms, together with substantial primary production by anoxygenic photosynthesis, also helps explain the limited net O$_2$ production by the cyanobacterial mat when measured in bulk (30). Overall, these findings emphasize the importance of microbial metabolic interactions in shaping biogeochemical processes in cyanobacterial mats under low-O$_2$ conditions, which dominated the long evolutionary history of cyanobacteria and played key but poorly understood roles in Earth's major geobiological turning points.

## MATERIALS AND METHODS

**Sample collection and sequencing.** This study used samples and metagenomic and metatranscriptomic sequence data produced as described by Voorhies et al. (92). Fifteen mat samples were collected by scuba divers from the R/V *Storm* between 2007 and 2012 from within a 100-m area of the Middle Island sinkhole (45.1984°N, 83.32721°W) by hand push cores of sediments, mat, and overlying groundwater (Table S1). Cores were rapidly transferred to the surface, and mats were separated from underlying sediments and submerged in RNAlater immediately shipboard. Less than 5 min elapsed between collection and preservation. Before preservation, mat samples were quickly washed with groundwater to remove as much sediment as possible. Mat structure ranged from conical structures we refer to as "fingers" (30) to prostrate mat. In May 2012, at the time of metagenomic and metatranscriptomic sequencing, conductivity, temperature, and dissolved O$_2$ in the overlying lake water as well as the groundwater above the mats were measured by a YSI 6600 multiparameter sonde.

DNA was extracted and processed for shotgun metagenomic sequencing as previously described (30). Samples were sequenced using an Illumina Hi Seq 2000 (paired end, 100 bp) instrument at the University of Michigan DNA Sequencing Core. In 2012, three samples of mat were collected at approximately 1 p.m. (day) and 1 a.m. (night) from within a 9-m$^2$ sampling area. RNA was extracted from these

six samples, randomly amplified with the MessageAmp II-Bacteria kit (Ambion), and converted to cDNA using the SuperScript double-stranded-cDNA synthesis kit (Invitrogen), as previously described (70). In the interest of cost efficiency and to minimize sample handling, rRNA was not removed (94, 95). cDNA was sequenced at the University of Michigan DNA Sequencing Core on an Illumina Hi Seq 2000 instrument producing paired-end reads.

**Assembly and genomic analysis.** In order to optimize assembly of genomes from low abundance members, a total of 922 million sequence reads from all 15 genomic DNA samples were combined and coassembled and binned by two different methods. We used IDBA-UD (96) for assembly and checked the results against previous assemblies of MIS mats that used multiple sequencing platforms and assembly programs, including a previously published metagenome based on 454 (28, 30), and assemblies of Illumina data from individual samples performed with Velvet (97). Specifically, the assembly was checked by verifying recovery of key genes and MAGs that were observed in the previous assemblies and by manual curation using the Integrated Genome Viewer (98) and Geneious (99) to visualize reads mapped to contigs and genes by BWA (Burrows-Wheeler aligner) (100) and look for signs of misassembly (e.g., discontinuities in coverage). Multiple strategies and software were used to generate metagenome-assembled-genome (MAG) bins. The first strategy used CONCOCT (101) to automate binning by differential coverage and tetranucleotide frequency for the subset of contigs that were 5 kb and larger. The resulting bins were refined manually in anvi'o (102, 103) and assigned taxonomy via Centrifuge (104) and CheckM (105). Likely due to high coverage and putative strain heterogeneity (37, 38), the 12.5-Mbp MAG bin representing the dominant cyanobacterium *Phormidium* had high completion (95.0%) but poor contamination (97.1%) metrics (106) (Table S2). For this bin as well as 6 other cyanobacterial MAG bins from the initial refinement, 6,011 contigs that were previously unbinned due to their short length (1,000 bp to 4,999 bp) were assigned to bins on the basis of similar coverage, nucleotide composition (tetranucleotide frequencies), and taxonomy via manual refinement in anvi'o.

We also employed a second, purely automated binning strategy for comparison. EukRep (107) removed eukaryotic contigs from the data set, and MetaBAT (108) used differential coverage and tetra-nucleotide frequency to generate MAGs from contigs 1,500 bp and longer. We again used CheckM to taxonomically identify MAG bins and tracked 16S rRNA, *psbA*, and *sqr* genes from the previously extracted cyanobacterial bins to identify their counterparts in the MetaBAT bins. Contigs previously assigned to *Phormidium* were poorly binned in this method. dRep (109) was used to pick the best representative bin from the two methods. Though 10 of the bins from CONCOCT+anvi'o were retained through dRep, the MetaBAT-generated bins were more often picked because they generally had lower estimates of contamination and strain heterogeneity. Putative *Phormidium* scaffolds that were not binned by MetaBAT were manually evaluated in anvi'o, and retained as the representative *Phormidium* bin in this analysis. Gene calling and functional annotation was performed by the Joint Genome Institute's Integrated Microbial Genomes Expert Review portal (https://img.jgi.doe.gov/cgi-bin/mer/main.cgi) (110).

Coverage of contigs by cDNA and DNA sequence reads from each sample was assessed by mapping reads to contigs using BWA (100) with default settings. Raw counts of cDNA reads (referred to here as counts) for each predicted protein-coding gene were determined using the IMG-derived coordinates of gene start and stop sites, along with the mapping information. rRNA genes erroneously called as protein-coding genes were identified by BLASTn against the SILVA SSU and LSU database, release 123 (111), and removed. The python script HTSeq.scripts.count from HTSeq (112) extracted transcript counts that unambiguously mapped to genes. In targeted searches for metabolic genes of interest, we identified 20 partial "genes" that were not suitable lengths when IMG-determined start and stop sites were used and were of appropriate lengths when partial "genes" were incorporated immediately upstream or downstream. Thus, for metatranscriptomic analyses, the counts of these partial genes were merged.

For analysis of metatranscriptomic data, only genes with at least two counts were considered. Two different normalization methods were used to analyze the metatranscriptomic data, depending on the question. First, transcript abundance was normalized by total mRNA reads recovered in each sample to calculate relative abundance of transcripts at the gene level. This metric is a function of both the organism abundance and expression per gene copy and provides a measure of total contribution to the transcript pool for each gene. Second, to compare relative gene expression within genomic bins (and remove the effect of dynamic community-wide transcript and organism abundance), we normalized relative abundance of transcripts by number of mRNAs mapped to each genome bin. To account for variability in sequencing effort between samples and for the impact of gene and read lengths, gene expression levels were normalized using TPM (113). To evaluate differences in expression levels between organisms of assembled metagenomic bins, TPM for bin-specific genes were also calculated with the denominator consisting of only reads recruited to the bin of interest. Statistical testing was conducted on the sample- and bin-normalized TPM of genes in RStudio using Kruskal-Wallis nonparametric tests and paired *t* tests, corrected with a Benjamini-Hochberg false discovery rate (*q*) of 0.05.

**Data availability.** Sequences from this study are available from NCBI under BioProject no. PRJNA72255. Reads from all 15 metagenomes and 6 metatranscriptomes are available in NCBI's Sequence Read Archive (Table S1). Accession numbers for MAGs that passed NCBI quality filtering are provided in Table S2.

## SUPPLEMENTAL MATERIAL

Supplemental material is available online only.

**FIG S1**, PDF file, 0.3 MB.

**FIG S2**, PDF file, 0.2 MB.

**FIG S3**, PDF file, 0.1 MB.
**FIG S4**, PDF file, 0.3 MB.
**FIG S5**, PDF file, 0.2 MB.
**FIG S6**, PDF file, 0.1 MB.
**FIG S7**, PDF file, 0.1 MB.
**FIG S8**, PDF file, 0.1 MB.
**TABLE S1**, PDF file, 0.1 MB.
**TABLE S2**, XLSX file, 0.03 MB.

## ACKNOWLEDGMENTS

Divers and crew of the *R/V Storm* NOAA and the Thunder Bay National Marine Sanctuary provided critical dive support and assistance with sampling and ship time. We thank the University of Michigan DNA sequencing core for DNA sequencing.

This work was supported by NSF grants EAR1035955 and EAR 1637066 to G.J.D., NSF grant EAR1035957 to B.A.B., the University of Michigan CCMB Pilot Grant to G.J.D., and the Scott Turner Award to A.A.V.

We declare no conflicts of interest.

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
