## [Reviewer comments · mSystems]

Omics-inferred partitioning and expression of diverse biogeochemical functions in a low-O₂ cyanobacterial mat community

Sharon Grim, Alexander Voorhies, Bopaiah Biddanda, Sunit Jain, Stephen C. Nold, Russ Green, and Gregory Dick

Corresponding Author(s): Gregory Dick, University of Michigan-Ann Arbor

Review Timeline:

Submission Date:	August 17, 2021
Editorial Decision:	September 13, 2021
Revision Received:	November 12, 2021
Accepted:	November 15, 2021

Editor: Hans Bernstein

Reviewer(s): The reviewers have opted to remain anonymous.

Transaction Report:

DOI: <https://doi.org/10.1128/mSystems.01042-21>

September 13, 2021

Prof. Gregory J Dick
University of Michigan-Ann Arbor
Dept. of Earth and Environmental Sciences
2534 North University Building
1100 N. University Ave.
Ann Arbor, MI 48109-1005

Re: mSystems01042-21 (Omics-inferred partitioning and expression of diverse biogeochemical functions in a low-O₂ cyanobacterial mat community)

Dear Prof. Gregory J Dick:

Thank you for submitting your manuscript to mSystems. We have completed our review and I am pleased to inform you that, in principle, we expect to accept it for publication in mSystems. However, acceptance will not be final until you have adequately addressed the reviewer comments.

Preparing Revision Guidelines

Sincerely,

Hans Bernstein

Editor, mSystems

Journals Department
Reviewer comments:

Reviewer #1 (Comments for the Author):

The manuscript (Omics-inferred partitioning and expression of diverse biogeochemical functions in a low-O₂ cyanobacterial mat community) presented by Grim, Voorhies, et al. offers a presentation and interpretation of metagenome and transcriptome data collected from an anoxic mat community. Overall, the manuscript is well written, provides a reasonable assessment of the data, and fills a relevant scientific gap in the literature. Overall, the manuscript reads very well, and I list below some of the very few concerns I have regarding this work:

- There is scant information provided on the actual sample collection other than it was located within the Middle Island Sinkhole and that bulk samples were collected. Additional information on the timing of these samples could be very useful for helping to unravel any seasonal patterns in the data. Presumably with 15 samples collected at seven timepoints over the 2007-2012 timespan there could be different seasons captured in the data.
- Similar to the comment above, there is some reference to previous work in this location, but very little biogeochemical data presented linked to the samples collected. For instance, transcriptomic samples were collected at 1pm and 1am to capture day and night conditions - what were the PAR levels during the day? Further, can the authors provide any insights to conditions like the O₂ gradient at the time of their sample collection, temperature, major dissolved components (for instance, how high was the methane loading), etc. Were these samples collected in conjunction with those being reported in the referenced previous work and therefore collected from the same set of conditions?
- Lines 97-99: "little is known about how anoxygenic photosynthesis by cyanobacteria operates at the transcriptomic level in complex.....". This statement is made a bit unclear by the term "operates at the transcriptomic level". Do the authors intend to imply that "little is known about transcriptomic controls on anoxygenic photosynthesis within cyanobacterial mats"?
- Lines 254-256: The authors may want to provide slightly more context as to why the high abundance of sulfate reducers implies they are found in the same mat regions as the cyanobacteria as it may be unclear to some readers.
- The authors provide an interesting assessment of physiological changes within the mat system resulting from diel cycling. Depicting relevant shifts in enzymatic activity, spatial location of cells, biogeochemical gradients, etc. resulting from diel shifts in a schematic may provide valuable context for some readers.

I enjoyed reading this manuscript and believe it will appeal to a fairly large readership.

Reviewer #2 (Comments for the Author):

Grim et al examined cyanobacterial mats that persist under low oxygen conditions at the sediment-water interface in Middle Island Sinkhole in Lake Huron by generating metagenomes and metatranscriptomes (contrasting day and night) from the mat. The metagenomic sequencing resulted in a number of metagenome assembled genomes including the most abundant cyanobacteria in the mat and other bacteria presumably involved in sulfur cycling. The data are complementary to a number of previous studies in the Middle Island Sinkhole, bolstering previous studies while also expanding current understanding of the system. RNA sequencing revealed that photosystem I transcripts were more abundant than photosystem II. The results have interesting implications for mats that persisted under low oxygen conditions on early Earth - for example, the potential for cyanobacterial photosystem I-driven primary productivity under low oxygen, sulfidic conditions. While anoxygenic photosynthesis via cyanobacteria has been inferred and potentially observed in other systems, linking cyanobacteria transcripts to high rates of anoxygenic photosynthesis previously observed in the sinkhole is particularly exciting.

In general the paper is well-written, the data analyses and interpretation are sufficient and consistent with the data presented. My one criticism of the results and discussion is that this section could include more quantitative details - the reader is tasked with looking at the figures for specifics. Some examples where numbers could be added: "Many of the *psaA* and *psbA* gene transcripts were more abundant at night than in the day", "the *Phormidium* SQR had by far the highest abundance of transcripts", "Transcripts of many *dsrA* genes were observed", "RuBisCo showed by far the highest transcript abundance of any autotrophic pathway"

My other main criticism of the paper is that the authors refer to significant differences regularly without detailing how significance was assessed. This includes both in the coverage data and transcript abundance. Perhaps some of these analyses were done within DESeq2 but these details are not provided.

Line 148: How was the assembly checked against previous assemblies? Is this what is described in lines 150-153?

Lines 160-163: I'm not sure I'm completely understanding the refinement strategy described here. You added contigs from the metagenome assembly that were between 1000 and 4999 bp in length to cyanobacterial bins that had high completion and poor contamination? How did adding short contigs to the bins change / improve them?

Line 295: Below you are careful to mention that transcripts do not equal protein so probably best to stick with transcript abundance here (rather than "gene expression").

Line 266: Could this also impact sulfide concentration (via sulfide oxidation?) and thus the balance of oxygenic vs. anoxygenic photosynthesis? Or perhaps there is a steady supply, etc (maybe not enough information / too speculative).

Line 368: Does this mean transcripts were abundant or different mores of dsrA (or both)?

Figure 1: Should there be a legend for bubble size? The pink is also very hard to see - perhaps a different symbol would be easier to discern.

Figure 6: Is it necessary to show all of the data here? It seems like you pick a threshold and only display data above that threshold.

Response to Reviewers

We thank the reviewers for their thoughtful and constructive comments on our manuscript. We have carefully considered these comments and thoroughly revised the manuscript accordingly. Please see our line-by-line response (**bold**) to reviewer comments (*italics*) below. Line numbers refer to the revised manuscript text file without markup.

Reviewer #1 (Comments for the Author):

The manuscript (Omics-inferred partitioning and expression of diverse biogeochemical functions in a low-O₂ cyanobacterial mat community) presented by Grim, Voorhies, et al. offers a presentation and interpretation of metagenome and transcriptome data collected from an anoxic mat community. Overall, the manuscript is well written, provides a reasonable assessment of the data, and fills a relevant scientific gap in the literature. Overall, the manuscript reads very well, and I list below some of the very few concerns I have regarding this work:

We thank the reviewer for this positive feedback.

• There is scant information provided on the actual sample collection other than it was located within the Middle Island Sinkhole and that bulk samples were collected. Additional information on the timing of these samples could be very useful for helping to unravel any seasonal patterns in the data. Presumably with 15 samples collected at seven timepoints over the 2007-2012 timespan there could be different seasons captured in the data.

We agree with the reviewer and have added the date of sampling to Table S1. The reviewer is correct that there are interesting seasonal trends in the sinkhole mat community. While the results described in the current manuscript are not sufficient for this purpose, we have another manuscript in preparation that is focused on this topic.

• Similar to the comment above, there is some reference to previous work in this location, but very little biogeochemical data presented linked to the samples collected. For instance, transcriptomic samples were collected at 1pm and 1am to capture day and night conditions - what were the PAR levels during the day? Further, can the authors provide any insights to conditions like the O₂ gradient at the time of their sample collection, temperature, major dissolved components (for instance, how high was the methane loading), etc. Were these samples collected in conjunction with those being reported in the referenced previous work and therefore collected from the same set of conditions?

We added a section on environmental setting and conditions to the beginning of the Results and Discussion section (L214-220):

“The environmental and geological setting of the Middle Island Sinkhole has been described previously (28). In May 2012, at the time of collection of samples for metatranscriptomic and metagenomic sequencing (Table S1), the groundwater layer ~1 m immediately above the mat in the sinkhole had substantially elevated specific conductivity (1813 $\mu\text{S cm}^{-1}$ vs 226 $\mu\text{S cm}^{-1}$ compared to the ambient lake water), lower and temporally consistent temperature (7-9°C), and an average of dissolved O₂ 3.37 mg L⁻¹.”

A short sentence describing the procedures for collecting this data was added to the Methods section (see L131-134):

“In May, 2012, at the time of metagenomic and metatranscriptomic sequencing, conductivity, temperature, and dissolved O₂ in the overlying lake water as well as the groundwater above the mats were measured by a YSI Multiparameter Sonde 6600.”

The results described in the current study are derived from samples that were described in Voorhies et al. 2015, which focused on viruses in the sinkhole community. We regret that additional biogeochemical and environmental data is not available.

• Lines 97-99: "little is known about how anoxygenic photosynthesis by cyanobacteria operates at the transcriptomic level in complex.....". This statement is made a bit unclear by the term "operates at the transcriptomic level". Do the authors intend to imply that "little is known about transcriptomic controls on anoxygenic photosynthesis within cyanobacterial mats"?

We have clarified this sentence as suggested by the reviewer, “...little is known about transcriptomic controls on cyanobacterial anoxygenic photosynthesis within cyanobacterial mats” (see L95-97)

• Lines 254-256: The authors may want to provide slightly more context as to why the high abundance of sulfate reducers implies they are found in the same mat regions as the cyanobacteria as it may be unclear to some readers.

We did not mean to differentiate different regions of the mat but rather that the SRBs were in the mat itself rather than just underlying sediments. We have clarified this sentence as follows (now L263-266):

“Based on their relatively high abundance, we infer that these sulfate-reducing bacteria were present directly in the cyanobacterial mat (66, 67) rather than the alternative that the sequences could be due to contamination of the mat by underlying sediments.”

• The authors provide an interesting assessment of physiological changes within the mat system resulting from diel cycling. Depicting relevant shifts in enzymatic activity, spatial location of cells, biogeochemical gradients, etc. resulting from diel shifts in a schematic may provide valuable context for some readers.

I enjoyed reading this manuscript and believe it will appeal to a fairly large readership.

We appreciate the reviewer’s thoughtful and supportive comments.

Reviewer #2 (Comments for the Author):

Grim et al examined cyanobacterial mats that persist under low oxygen conditions at the sediment-water interface in Middle Island Sinkhole in Lake Huron by generating metagenomes

and metatranscriptomes (contrasting day and night) from the mat. The metagenomic sequencing resulted in a number of metagenome assembled genomes including the most abundant cyanobacteria in the mat and other bacteria presumably involved in sulfur cycling. The data are complementary to a number of previous studies in the Middle Island Sinkhole, bolstering previous studies while also expanding current understanding of the system. RNA sequencing revealed that photosystem I transcripts were more abundant than photosystem II. The results have interesting implications for mats that persisted under low oxygen conditions on early Earth - for example, the potential for cyanobacterial photosystem I-driven primary productivity under low oxygen, sulfidic conditions. While anoxygenic photosynthesis via cyanobacteria has been inferred and potentially observed in other systems, linking cyanobacteria transcripts to high rates of anoxygenic photosynthesis previously observed in the sinkhole is particularly exciting.

*In general the paper is well-written, the data analyses and interpretation are sufficient and consistent with the data presented. My one criticism of the results and discussion is that this section could include more quantitative details - the reader is tasked with looking at the figures for specifics. Some examples where numbers could be added: "Many of the *psaA* and *psbA* gene transcripts were more abundant at night than in the day", "the *Phormidium* SQR had by far the highest abundance of transcripts", "Transcripts of many *dsrA* genes were observed", "RuBisCo showed by far the highest transcript abundance of any autotrophic pathway"*

We appreciate the reviewer's suggestion to add numbers and have done so as follows:

- 1. Edit: "21 of 32 *psaA* and *psbA* genes had transcripts that were more abundant at night than in the day, although these differences were not statistically significant (Fig. 2). When normalized by the number of transcript reads recruited to each bin, which removes effects of transcriptomic variability across the whole community on transcript counts for genes within each bin, 9 of 21 genes had transcripts that were more abundant at night (Fig. S4)." (L316-320)**
- 2. Removed "by far" from the following (we feel that additional numbers are not helpful in this case): "Of the five cyanobacterial *sqr* homologs recovered from the MIS community, the *Phormidium* SQR had the highest abundance of transcripts, from both its type I and II *sqr* genes (Fig. 4)." (L360-362)**
- 3. Edit: "Transcripts of seven different *dsrA* genes were observed..." (L382-384)**
- 4. Edit: "RuBisCo had the highest transcript abundance of any autotrophic pathway." (L412-413)**

My other main criticism of the paper is that the authors refer to significant differences regularly without detailing how significance was assessed. This includes both in the coverage data and transcript abundance. Perhaps some of these analyses were done within DESeq2 but these details are not provided.

We made the following revisions to clarify how statistical significance was assessed and where differences were statistically significant and where they were not:

- 1. Removed "significant" from L43 of the abstract.**

2. The reviewer's comment on DESeq2 made us realize that these methods had been erroneously retained after we had removed results from DESeq2 from a previous version of the manuscript that focused on comparison of community-wide transcript abundance patterns. Thus, we removed the two sentences from the methods section that referenced DESeq2 (L190, 195) and added a sentence describing how the statistical analyses were done (L203-206): "Statistical testing was conducted on the sample- and bin-normalized TPM of genes in RStudio using Kruskal-Wallis nonparametric tests and paired t-tests, corrected with Benjamini-Hochberg false discovery rate of $q = 0.05$."
3. We clarified that there was no significant difference between abundance of transcripts in day vs. night (L316-317): "21 of 32 *psaA* and *psbA* genes had transcripts that were more abundant at night than in the day, although these differences were not statistically significant (Fig. 2)."
4. We clarified that there was no statistically significant difference between ratio of transcript abundance for PSII:PSI genes (L343-345): "However, likely due to high variability of the abundance of diatom transcripts for these genes, this large difference in ratio of transcript abundance from PSII:PSI genes was not statistically significant." We also noted that it was not useful for us to include the unassigned transcript counts (which come from different and unknown organisms) in figure 3, so the unassigned panel was removed.
5. We clarified that there was a statistical difference in transcript abundance of the different *sqr* genes (L362-363): "The bin-normalized TPM of *Phormidium*'s *sqrII* was significantly higher than *sqrI* ($p < 0.05$)."

Line 148: How was the assembly checked against previous assemblies? Is this what is described in lines 150-153?

We thank the reviewer for pointing out the need to clarify this passage, which we did by rewriting as follows (now L148-155): "We used IDBA-UD (40) for assembly, and checked it against previous assemblies of MIS mats that used multiple sequencing platforms and assembly programs, including a previously published metagenome based on 454 (28, 30), and assemblies of Illumina data from individual samples performed with Velvet (41). Specifically, the assembly was checked by verifying recovery of key genes and MAGs that were observed in the previous assemblies and by manual curation using the Integrated Genome Viewer (42) and Geneious (43) to visualize reads mapped to contigs and genes by BWA (44) and look for signs of mis-assembly (e.g., discontinuities in coverage)."

Lines 160-163: I'm not sure I'm completely understanding the refinement strategy described here. You added contigs from the metagenome assembly that were between 1000 and 4999 bp in length to cyanobacterial bins that had high completion and poor contamination? How did adding short contigs to the bins change / improve them?

Of the 7105 cyanobacterial contigs that had been excluded from binning solely due to length < 5000 bp, the refinement strategy was able to assign 6011 contigs to bins. These contigs included genes with core metabolic genes with potentially >1 copy number such as photosynthetic reaction center gene *psbA*, thus they improved the bins both

quantitatively and qualitatively. We have clarified these details by rewriting this passage as follows (now L162-165): “For this bin as well as 6 other cyanobacterial MAG bins from the initial refinement, 6,011 contigs that were previously unbinned due to their short length (1000bp – 4999bp) were assigned to bins on the basis of similar coverage, nucleotide composition (tetranucleotide frequencies), and taxonomy via manual refinement in anvi’o.”

Line 295: Below you are careful to mention that transcripts do not equal protein so probably best to stick with transcript abundance here (rather than "gene expression").

We agree with the reviewer and changed this to “The most abundant transcripts for *psaA* and *psbA* genes were from...” (now L305-307).

Line 266: Could this also impact sulfide concentration (via sulfide oxidation?) and thus the balance of oxygenic vs. anoxygenic photosynthesis? Or perhaps there is a steady supply, etc (maybe not enough information / too speculative).

We thank the reviewer for this interesting point. While it is unclear to what extent lithotrophic sulfide oxidation influences the balance of oxygenic vs. anoxygenic photosynthesis, we feel that it is appropriate to state the following (now L277-279): “These large sulfur-oxidizing bacteria likely contribute to the substantial rates of chemosynthesis measured previously (30) and likely influence cyanobacterial photosynthesis by consuming sulfide.”

*Line 368: Does this mean transcripts were abundant or different mores of *dsrA* (or both)?*

We clarified that this was referring to the number of different *dsrA* genes (L382-384): “Transcripts of seven different *dsrA* genes were observed, and the presence of *dsrD* on the same scaffold (Fig. S7) was used to confirm inclusion of these genes in the dissimilatory sulfite reductase pathway (*dsr*).”

Figure 1: Should there be a legend for bubble size? The pink is also very hard to see - perhaps a different symbol would be easier to discern.

Bubble size is not meaningful in this figure. To address the concern about visibility of the pink circles, we changed it to an “X” (see below).

Figure 6: Is it necessary to show all of the data here? It seems like you pick a threshold and only display data above that threshold.

This figure has two purposes: (i) to show the diversity of autotrophy genes recovered from the MAGs, and (ii) to indicate their relative levels of transcriptional activity. In the interest of (i), we feel it is important to show even the genes that have low transcriptional activity, thus we propose to keep the figure in its current form.

November 15, 2021

Prof. Gregory J Dick
University of Michigan-Ann Arbor
Dept. of Earth and Environmental Sciences
2534 North University Building
1100 N. University Ave.
Ann Arbor, MI 48109-1005

Re: mSystems01042-21R1 (Omics-inferred partitioning and expression of diverse biogeochemical functions in a low-O₂ cyanobacterial mat community)

Dear Prof. Gregory J Dick:

Your manuscript has been accepted, and I am forwarding it to the ASM Journals Department for publication. For your reference, ASM Journals' address is given below. Before it can be scheduled for publication, your manuscript will be checked by the mSystems senior production editor, Ellie Ghatineh, to make sure that all elements meet the technical requirements for publication. She will contact you if anything needs to be revised before copyediting and production can begin. Otherwise, you will be notified when your proofs are ready to be viewed.

Publication Fees:

We recognize that the video files can become quite large, and so to avoid quality loss ASM suggests sending the video file via <https://www.wetransfer.com/>. When you have a final version of the video and the still ready to share, please send it to Ellie Ghatineh at eghatineh@asmusa.org.

Sincerely,

Hans Bernstein
Editor, mSystems

Journals Department
Fig. S1: Accept
Fig. S6: Accept
Fig. S5: Accept
Fig. S7: Accept
Fig. S3: Accept
Table S2: Accept
Fig. S4: Accept
Fig. S2: Accept
Table S1: Accept
Fig. S8: Accept